



# Basal Water Storage Variations beneath Antarctic Ice Sheet Inferred from Multi-source Satellite Data

Jingyu Kang[1,2], Yang Lu[1], Yan Li[1,2], Zizhan Zhang[1] and Hongling Shi[1]

[1]State Key Laboratory of Geodesy and Earth's Dynamics, Innovation Academy for Precision Measurement Science and Technology, Wuhan 430071, China.
[2]University of Chinese Academy of Sciences, Beijing 100049, China

*Correspondence to*: Jingyu Kang (kangjingyu17@mails.ucas.ac.cn)

**Abstract** Antarctic basal water storage variations (BWSV) refer to the mass variations of liquid water beneath Antarctic ice sheet. Identifying these variations is critical to understand the behaviour of ice sheet, yet it is rarely accessible to direct observation. We presented a layered gravity density forward/inversion method for estimating Antarctic BWSV from multi-source satellite observation data, and relevant models. Results reveal spatial variability of BWSV with the mean rate of 43±13 Gt/yr during 2003-2009, which is 21 Gt/yr lower than basal melting rate. This indicates that the basal meltwater beneath Antarctic ice sheet is decreasing with the rate of -21±13 Gt/yr, accounting for 28% of the mass balance rate (-76 Gt/yr, Shepherd et al. (2018)), and the basal water migrations between basal drainage systems and oceans is non-ignorable in estimating basal mass balance of Antarctic ice sheet. Similar spatial distribution of basal water increases regions and locations of active subglacial lakes indicates that basal water storage in most active subglacial lakes is increasing. In most region of Antarctic ice sheet except Amundsen Sea coast region, the comparison of spatial BWSV and ice velocity displays a positive correlation between considerable basal water increases and rapid/accelerated ice flows, which indicates that BWSV appear to have an important effect on ice flows. Accordingly, we infer that further enhanced flow velocities are expected if basal water continues to increase in these regions.

## 1. Introduction

Antarctic basal water is widely generated by geothermal heating, basal pressure melting and frictional heating. The basal liquid water converge on subglacial lakes or spread over ice-bed interface, and connected to each other (Wingham et al., 2006;Fricker et al., 2016), which forms basal drainage systems with complex basal water migrations (Pattyn, 2008;Carter et al., 2015). Temporal mass variations in Antarctic basal liquid water storage are called basal water storage variations (BWSV), it influences basal effective pressure and trigger changing ice sheet velocities (Bell and Robin, 2008;Fricker et al., 2007;Alley, 1992).

BWSV are mainly controlled by basal melting rate and basal water migrations. In ice-bed interface, basal ice melt in high basal melting rate regions to increase liquid water storage, the liquid water refreezes when flowing through supercooling regions to





reduce the storage, besides, basal water migrations between basal drainage systems and oceans also influence spatial basal
water storage variation. To date, many studies have been conducted on Antarctic basal condition (Alley et al., 1998;Rignot
and Jacobs, 2002;Augustin et al., 2007;Fisher et al., 2015;Martos et al., 2017;Liefferinge et al., 2018), particularly, Pattyn
(2010) inferred the Antarctic basal melting rate using a hybrid ice sheet/ice streams model. Consequently, a key question in
exploring BWSV is to evaluate the basal mass balance (BMB) that is mainly caused by basal water migrations.

Although studies have been performed on the surface ice sheet and ice shelves, BMB over the ice sheet remains poorly
understood due to limited observation. The commonly adopted approach is to examine the local surface height variations
provided by altimetry observations, by assuming surface height variations related to subglacial lakes discharge (Flament et al.,
2014;Wingham et al., 2006). However, this method may be invalid if BMB occurs in regions without sufficient surface
expressions or with complex basal conditions, such as the Siple Coast (Young et al., 2016). Göeller et al. (2012) proposed a
balanced water layer concept in large scale ice sheet models to present BMB, while the modelled results depend largely on the
reliability of data for building models.

In this paper, gravity variations over Antarctic ice sheet (AIS) are considered as a combination of the gravity variations caused
by surface mass redistribution, basal mass balance (BMB), glacial isostatic adjustment (GIA), and ice sheet's vertical
movement. We adopted an elaborate gravity forward modelling and surface density discrimination method in combination
with Gravity Recovery and Climate Experiment (GRACE) data, ICESat data, GPS data, and relevant models to separate the
initial gravity variations caused by BMB, and employed a layered gravity density inversion method was to estimate the
equivalent water height (EWH) of BMB. Afterward, the EWH of BMB was used to recalculate the gravity variation caused by
surface mass redistribution and ice sheet's vertical movement, by assuming that BMB can be expressed on surface height
variation, until the BMB result is stable. Finally, we combined BMB with basal melting rate data to evaluate BWSV. The
results were verified and interpreted based on existing research.

## 2.   Methods

### 2.1 Temporal variations in gravity field and surface height on Antarctic ice sheet

Active mass redistributions on Antarctic ice sheet (AIS) occur mainly in the surface of ice sheet, ice-bed interface, and solid
Earth (Fig. 1). In the surface of AIS, mass redistributions are dominated by firn layer's processes of snow accumulation,
sublimation/runoff, melting and firn compaction (Ligtenberg, 2014), and ice flows. In ice-bed interface, mass redistributions
refer to basal mass balance (BMB) that is mainly caused by basal water migrations between basal drainage systems and oceans.
Mass redistributions of solid Earth mainly refer to glacial isostatic adjustment (GIA) that describes Earth's isostatic
deformation since the last glacial stage, such as mantle convection, and Earth's load deformation caused by load of the
overlying ice sheet.



Mass redistributions lead to temporal variations in gravity field defined in geoid, and can be captured by Gravity Recovery and Climate Experiment (GRACE) observations. These variations contain surface gravity variations caused by snow accumulation, sublimation/runoff and ice flow, gravity variations of BMB in ice-bed interface, and gravity variations of solid Earth that arise from the combined effect of GIA and Earth's load deformation. Besides, BMB and basal melting/refreezing also influence ice sheet's vertical movement by controlling basal mass changes and ice-water phase change process, thereby

contributing to gravity variations. Accordingly, we can express the Antarctic integrated time-variable gravity field as a superposition of layered gravity variations:

$$\Delta dg_{GRACE} = \Delta dg_{surf} + \Delta dg_{BMB} + \Delta dg_{GIA} + \Delta dg_{IVM} \tag{1}$$

where $\Delta dg_{GRACE}$ are Antarctic integrated time-variable gravity variations (including load effect) observed by GRACE (section 3.1); $\Delta dg_{surf}$ are gravity variations caused by Antarctic surface mass redistributions process, including firn layer's processes

and ice flow; $\Delta dg_{BMB}$ are BMB-derived gravity variations; $\Delta dg_{GIA}$ are gravity variations caused by Earth's isostatic deformation and mass changes in the interior of the Earth, and can be obtained from GIA model (section 3.4); $\Delta dg_{IVM}$ are additional gravity variation caused by ice sheet's vertical movement.

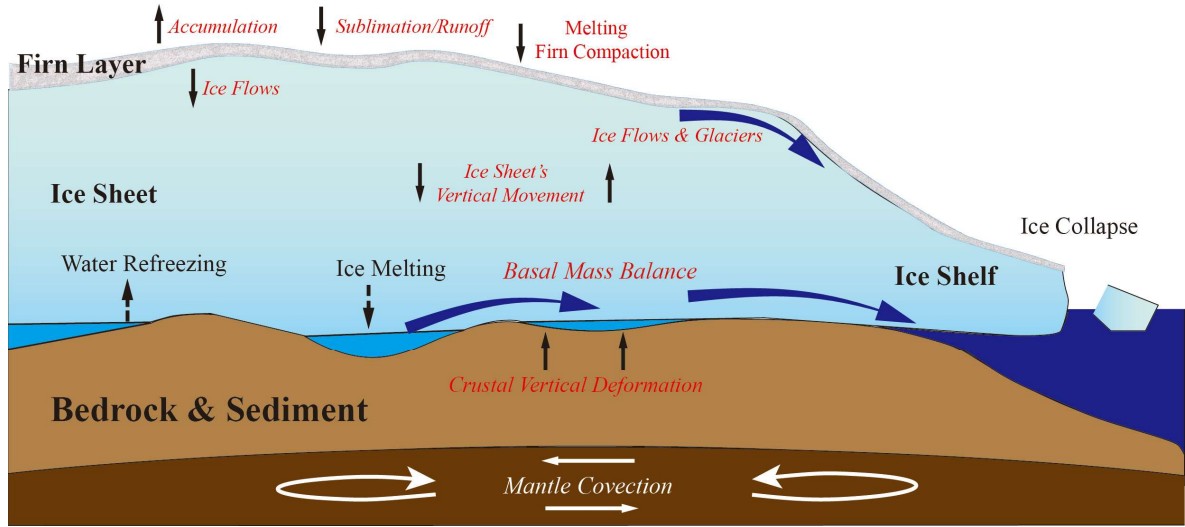

**Figure 1** Mass redistributions in different layers on AIS. Italics are components that contribute to gravity variations, and red fonts refer to components resulting in surface height variations. Ice sheet's vertical movement is controlled by combined effect of BMB, basal water refreezing and ice melting. Crustal vertical deformation refers to Earth's isostatic and load deformation.

Mass redistributions also lead to surface height variations of AIS:

$$dh_{ICESat} = dh_{FL} + dh_{IF} + dh_{IVM} + dh_{CVD} \tag{2}$$

where $dh_{ICESat}$ are AIS surface height variations that can be observed by ICESat (section 3.2); $dh_{FL}$ are height variations caused by process of firn layer including surface snow accumulation, sublimation/runoff, melting and firn compaction. $dh_{IF}$





are height variations caused by ice flow; $dh_{IVM}$ are the ice sheet's vertical movement; $dh_{CVD}$ are crustal vertical deformations that be calculated by fitting GIA predicted basal uplift rate to sparse GPS observation rate (section 3.4).

**2.2 Estimation of basal mass balance and basal water storage variations**

Separation of $\Delta dg_{BMB}$ from integrated time-variable gravity field (Equation 3) is enabled by different sensitivity of satellite observation with respect to gravity and height variations in different layers. According to Equation 1, $\Delta dg_{BMB}$ can be expressed as follows:

$$\Delta dg_{BMB} = \Delta dg_{GRACE} - \Delta dg_{surf} - \Delta dg_{GIA} - \Delta dg_{IMV} \tag{3}$$

In Equation 3, $\Delta dg_{surf}$ and $\Delta dg_{IVM}$ can be calculated through gravity forward modelling method and corresponding density
data, providing the known height variations caused by surface process and ice sheet's vertical movement. Then mass changes of BMB can be estimated from $\Delta dg_{BMB}$ by utilizing gravity density inversion method. However, height variations caused by surface process are complicated, especially in ice flow regions where the surface height variations are dominated by ice thinning, and the combination of surface and ice density is more appropriate in gravity forward modelling process; furthermore, BMB and basal melting/refreezing cause ice sheet's vertical movement, and this movement is also expressed as height
variations and result in additional gravity variation $\Delta dg_{IVM}$. These coupled variations are contained in gravity and surface height variations that captured by satellite gravity and altimetry mission. Therefore, the key problem of separating $dg_{BMB}$ from integrated time-variable gravity field is to distinguish different height variation components on each layer and IVM from altimetry observation data. For this purpose, we adopted an iteration method that accounts for surface density discrimination and IVM correction, to estimate the corresponding height and gravity variation (Fig. 2), details are described as follows.

Firstly, we set the initial value of $dh_{IVM}$ to 0, and deducted $dh_{CVD}$ and $dh_{IVM}$ from AIS surface height variation $dh_{ICESat}$, to obtain the initial height variation from Antarctic surface mass redistributions $dh_{surf} = dh_{ICESat} - dh_{IVM} - dh_{CVD}$. It is noticed that $dh_{surf}$ only contains height variations caused by firn layer's processes $dh_{FL}$ and ice flow $dh_{IF}$. Afterward, we utilized surface density discrimination method (Gunter et al., 2014), in combination with a semi-empirical estimated time-dependent firn densification model (FDM) that describes spatio-temporal evolution of Antarctic firn layer (Ligtenberg et al.,
2011) (section 3.3), to discriminate height variation component from dh$_{surf}$. The corresponding surface density $\rho$ is assigned as follows:

$$\rho = \begin{cases} \rho_{firn}\, for\, dh_{FDM},\ \rho_{ice}\, for\, (dh_{surf} - dh_{FDM}) & if\ (dh_{surf} - dh_{FDM}) < 0\, \&\, |dh_{surf} - dh_{FDM}| > 2\sigma_{dh} \\ \rho_{surf}\, for\, dh_{surf} & if\ (dh_{surf} - dh_{FDM}) > 0\, \&\, |dh_{surf} - dh_{FDM}| > 2\sigma_{dh} \\ 0 & otherwise \end{cases} \tag{4}$$

where $\rho_{firn}$ are firn density distributions (Ligtenberg et al., 2011); $dh_{FDM}$ are height variations derived from FDM that provides temporal surface height variations caused by the surface mass balance variations, liquid water processes (snowmelt,





percolation, refreezing and runoff) and firn compaction (Ligtenberg et al., 2011). $\rho_{ice}$ is ice density (917 kg m$^{-3}$) and $\sigma_{dh} = \sqrt{\delta_{ICESat}^2 + \delta_{FDM}^2}$.

    As shown in Equation 4, the negative height differences between $dh_{surf}$ and $dh_{FDM}$ greater than $2\sigma_{dh}$ are attributed to ice flow (that is, $dh_{IF} = dh_{surf} - dh_{FDM}$) with the assignment of ice density $\rho_{ice}$, and surface firn density $\rho_{firn}$ is assigned to $dh_{FDM}$. Similarly, positive height difference greater than $2\sigma_{dh}$ are attributed to underestimation of FDM, and surface firn

density $\rho_{firn}$ is assigned to $dh_{surf}$. Then gravity forward modelling method (section 2.3) is employed to calculate $dg_{surf}$ in Equation 3.

    Secondly, we set initial $\Delta dg_{IVM}$ to 0, and deducted initial value of $\Delta dg_{surf}$, $\Delta dg_{GIA}$ and $\Delta dg_{IVM}$ from Antarctic integrated time-variable gravity $\Delta dg_{GRACE}$, to obtain the initial $\Delta dg_{BMB}$ in Equation 3. Then gravity inversion method (section 2.3) was used to estimate initial mass variations of BMB ($\Delta m_{BMB}$, in form of equivalent water height (EWH)), and a 300km gaussian

filter was performed on $\Delta m_{BMB}$ to match the spatial resolution of GRACE data. Thirdly, we used basal melting rate data (section 3.5) (Pattyn, 2010) to judge whether basal is in melting or freezing condition, and combined $\Delta m_{BMB}$ to calculate $dh_{IVM}$. The calculation of $dh_{IVM}$ was performed based on the assumption of ice layer's instant vertical movement in response to BMB: in basal melting regions, ice sheet's vertical movement $dh_{IVM}$ is replaced by $\Delta m_{BMB}$; in basal freezing regions, $dh_{IVM}$ was replaced by $\Delta m_{BMB} * \rho_{ice}/1000$. The ice layer's instant response assumption is reasonable and can be justified by strong

surface height variations (up to 90m/yr) caused by subglacial lakes water changes in a short time (Fricker et al., 2016). Afterward, $dh_{IVM}$ was used to calculate the gravity correction of ice sheet's vertical movement $\Delta dg_{IVM}$. Calculation of $\Delta dg_{IVM}$ can be achieved by estimating gravity difference between ice sheet before and after it moves, while this correction method is multifarious due to the complication of surface process. For simplicity, we only updated the value of $dh_{IVM}$ in first step, and repeated the surface density discrimination method for iteration, until the $\Delta m_{BMB}$ is stable (with the total BMB difference

between two iterations smaller than 5Gt/yr). Finally, we combined $\Delta m_{BMB}$ and basal melting rate data (Pattyn, 2010) to evaluate basal water storage variations ($\Delta m_{BWSV}$, in form of equivalent water height (EWH)): regions with basal melting rate greater than 0 are assumed to maintain basal water migrations, and the ($\Delta m_{BWSV}$ are the gross of $\Delta m_{BMB}$ and basal ice melting; regions with the melting rate of 0 are assumed to be at supercooling condition, and no BWSV occurs in these regions.




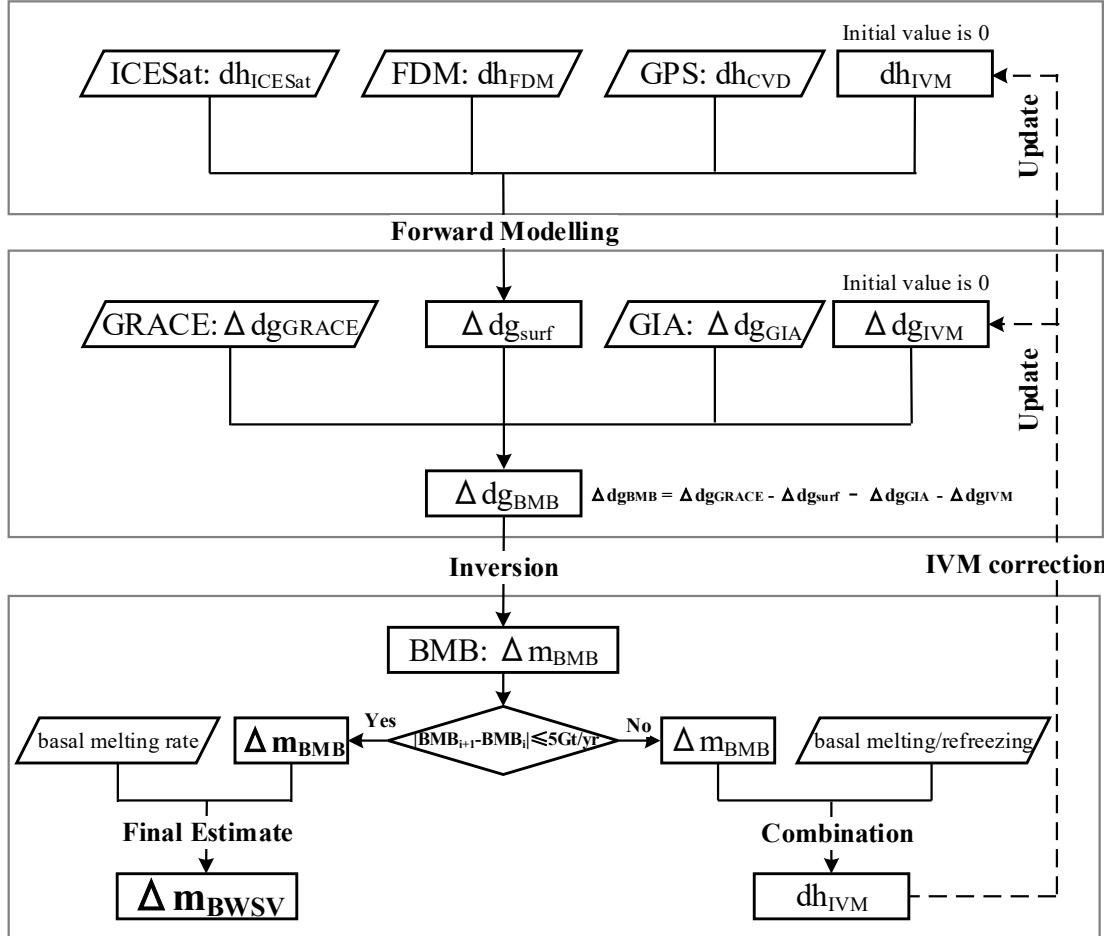

**Figure 2** Flowchart for estimating basal mass balance (BMB) and basal water storage variations (BWSV). Firstly, initial values of Antarctic surface gravity variations $\Delta dg_{surf}$ are calculated through gravity forward modelling method in combination with ICESat, FDM, GPS and GIA data. Secondly, BMB-derived gravity variations $\Delta dg_{BMB}$ are abstracted from GRACE, and BMB are estimated through gravity density inversion method. Thirdly, ice sheet's vertical movement (IVM) are used to perform gravity correction. For simplicity, IVM is only updated in first step for iteration in order to avoid multifarious gravity correction of $\Delta dg_{IMV}$.

### 2.3 Gravity density inversion method based on forward modelling

Gravity forward modelling is used to convert the volume variations and density distribution data into gravity variations. In this study, we adopted an oblique triangular prism that addresses terrain surfaces on ellipsoids without gaps. For this purpose, surface of AIS was divided into many equilateral triangles with sides approximately 50km (Fig. 3). Gravity variations are given by the basic gravity forward modelling Equation (Hofmann-Wellenhof and Moritz, 2006):



$$dg = -G\rho \iiint \frac{z}{(x^2+y^2+z^2)^{3/2}} dxdydz \qquad (5)$$

where $G$ is the gravitational constant, $\rho$ are density distributions related to volume variations (section 3.3).

Volume integral in Equation 5 can be modified into surface integral, accordingly, gravity variations components in different Antarctic layers are expressed as the sum of gravity variations of oblique triangular prism:

$$dg = \sum_{k=1}^{N} G\rho_k [\iint \frac{1}{(x_k^2+y_k^2+z_{k\_l}^2)^{\frac{1}{2}}} dx_k dy_k - \iint \frac{1}{(x_k^2+y_k^2+z_{k\_u}^2)^{\frac{1}{2}}} dx_k dy_k] \qquad (6)$$

where $dg$ are gravity variations in AIS layers that defined in geoid, $N$ is the number of triangular prisms within the integrated

radius (500km), $\rho_k$ are density of each triangular prisms, $z_{k\_l}$ and $z_{k\_u}$ are lower/upper surface height of each triangular prisms (that is, AIS's basal and surface height that comes from BEDMAP2 (Fretwell et al., 2013)).

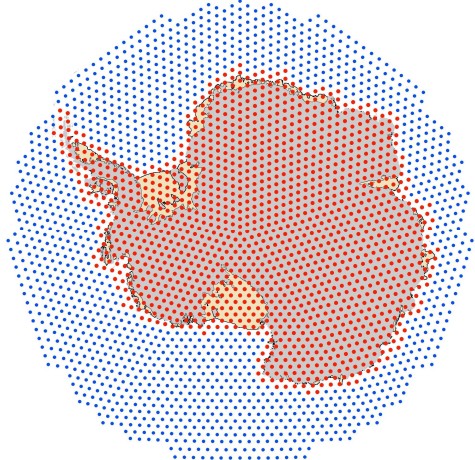

**Figure 3** Vertexes of triangles used for gravity forward modelling/inversion. Grey regions are Antarctic ice sheets (AIS), yellow regions are Antarctic ice shelves. Red points are vertexes of 10314 triangles used for estimating BMB through gravity inversion method, and these triangles are extended 1° outward for reducing possible leakage-out errors of gravity signals.

Then a layered gravity density inversion method was developed to estimate BMB. In this method, BMB was assumed to occur on a thin layer in Antarctic ice-bed interface, with a fixed thickness and variable density. Then BMB induced gravity variations can be expressed as follow:

$$\Delta dg_{BMB} = \sum_{k=1}^{N} G\rho_k [\iint \frac{1}{(x_k^2+y_k^2+z_k^2)^{\frac{1}{2}}} dx_k dy_k - \iint \frac{1}{(x_k^2+y_k^2+(z_k+dz)^2)^{\frac{1}{2}}} dx_k dy_k] \qquad (7)$$

$$\rho_k = \rho_w H_k \qquad (8)$$



where $dz$ is the thickness of the fixed layer on Antarctic bedrock, $\rho_k$ is fixed thin layer's density distribution, $\rho_w$ is water density, $H_k$ is the EWH of BMB that need to be estimated.

Then, Equation 7 could be modified as follows:

$$\frac{\Delta dg_{BMB}}{G\rho_w} = \sum_{k=1}^{N} H_k \left[ \iint \frac{1}{\left(x_k^2+y_k^2+z_k^2\right)^{\frac{1}{2}}} dx_k dy_k - \iint \frac{1}{\left(x_k^2+y_k^2+(z_k+dz)^2\right)^{\frac{1}{2}}} dx_k dy_k \right] \tag{9}$$

For M gravity points (M=10314), Equation 9 can be expressed as:

$$d = BX \tag{10}$$

where

$$d = \begin{bmatrix} \dfrac{\Delta dg_{BMB_1}}{G\rho_w} \\ \vdots \\ \dfrac{\Delta dg_{BMB_M}}{G\rho_w} \end{bmatrix}; \; B = \begin{bmatrix} T_{11} & \cdots & T_{1N} \\ \vdots & \ddots & \vdots \\ T_{M1} & \cdots & T_{MN} \end{bmatrix}; X = \begin{bmatrix} H_1 \\ \vdots \\ H_N \end{bmatrix};$$

$$T_{MN} = \left[ \iint \frac{1}{\left(x_{mn}^2+y_{mn}^2+z_{mn}^2\right)^{\frac{1}{2}}} dx_{mn} dy_{mn} - \iint \frac{1}{\left(x_{mn}^2+y_{mn}^2+(z_{mn}+dz)^2\right)^{\frac{1}{2}}} dx_{mn} dy_{mn} \right]$$

Then the conjugate gradient method was employed to solve the EWH of BMB (that is, X in Equation 10).

## 2.4 Uncertainty estimation

Estimating uncertainties of BMB/BWSV is comprehensive due to the combination of multi-source data in gravity forward/inversion including GRACE, ICESat, FDM, GPS, GIA, and basal melting rate. Among them, Uncertainties of GRACE are provided by calibrated errors for spherical harmonic coefficients, and can be estimated utilizing the method of Wahr et al. (2006); Others are provided by uncertainties of volume variations or uplift rate, and can be converted to associated gravity uncertainties by using cylinder model (Heiskanen and Moritz, 1967) combined with error propagation law. The uncertainty of BMB is obtained as follows:

$$\delta\Delta dg_{BMB} = \sqrt{\delta\Delta dg_{GRACE}^2 + \delta\Delta dg_{ICESat}^2 + \delta\Delta dg_{FDM}^2 + \delta\Delta dg_{GPS}^2 + \delta\Delta dg_{GIA}^2} \tag{11}$$

where $\delta\Delta dg_{GRACE}$, $\delta\Delta dg_{ICESat}$, $\delta\Delta dg_{FDM}$, $\delta\Delta dg_{GPS}$, $\delta\Delta dg_{GIA}$ are uncertainty of gravity variations related to GRACE, ICESat, FDM, GPS and GIA respectively.

Afterward, uncertainties of BMB $\delta\Delta m_{BMB}$ are calculated from $\delta\Delta dg_{BMB}$, based on the inversion of cylinder model and error propagation law, and uncertainties of BWSV $\delta\Delta m_{BWSV}$ are estimated as follow:

$$\delta\Delta m_{BWSV} = \sqrt{\delta\Delta m_{BMB}^2 + \delta\Delta m_{BM}^2} \tag{12}$$



where $\delta\Delta m_{BMB}$ refers to EWH uncertainties of BMB, $\delta\Delta m_{BM}$ refers to standard deviations of basal melting rate (Pattyn, 2010;Van Liefferinge and Pattyn, 2013).

## 3.    Data processing

### 3.1    Time-varying gravity data derived from GRACE

Time-varying gravity data comes from three Release 06 (RL06) monthly GRACE gravity field solutions provided by the
University of Texas Center for Space Research (CSR), the Jet Propulsion Laboratory (JPL), and the German Research Centre for Geosciences (GFZ). Each solution is represented by fully normalized Stokes potential coefficients with degree and order to 60. The C20 coefficients of each GRACE solution were replaced by satellite laser ranging (Cheng et al., 2013), and the degree 1 coefficients are inserted by the values of Swenson et al. (2008). Striping errors were eliminated by using P4M6 filter (Chen et al., 2007) and 300km Gaussian smoothing filter. Scaling factors was multiplied to restore the amplitude dampening
and leakage-out errors (Gao et al., 2015). In this study, basal mass balance signals contained in GRACE only occurs in AIS region, so leakage-in errors corrections were not performed here. Linear term was abstracted through least squares adjustment of six-parameter function consisting of constant term, linear trend, annual and semi-annual periodic signals. The gravity variation trend was calculated according to the spherical-harmonic expansion of the gravity anomaly (Heiskanen and Moritz, 1967) (Equation 13), and we used the mean gravity variations trend from the three GRACE solutions to account for time-
varying gravity on AIS. It is worth noting that time-varying gravity contains not only linear trend of AIS gravity variations, but also load effect. Then the AIS time-varying gravity trend $dg_{GRACE}$ is expressed as follows:

$$\Delta dg_{GRACE} = \frac{GM}{R^2}\sum_{n=2}^{max}(n-1)\frac{1}{1+k_n}\sum_{m=0}^{n}\bar{P}_{nm}(cos(\theta))[\Delta C_{nm}\,cos(m\phi) + \Delta S_{nm}\,sin(m\phi)] \quad (13)$$

where $GM$ is the geocentric gravitational constant, $R$ is the mean radius of Earth, $k_n$ is the load Love number of degree $n$ (Wahr et al., 1998), $\bar{P}_{nm}$ are the normalized associated Legendre functions, $\Delta C_{nm}$ and $\Delta S_{nm}$ are modified harmonic
coefficients, $\theta$ and $\phi$ are colatitude and longitude, and $r(\theta)$ is the geocentric distance related to $\theta$.

The uncertainties were estimated through the method of Wahr et al. (2006), from the calibrated errors for spherical harmonic coefficients provided by CSR, JPL and GFZ, and coefficient errors of degree 1 and degree 2 were replaced by the corresponding standard deviations provided by Cheng and Swenson (Cheng et al., 2013;Swenson et al., 2008). Figure 4a-4f show linear trend and corresponding uncertainties of AIS gravity variations respected to three GRACE solutions over the
period between February 2003 and October 2009, which is consistent with the ICESat simultaneously observation period in order to avoid sampling effects. All figures of AIS region are shown in Antarctic polar stereographic projection coordinates.



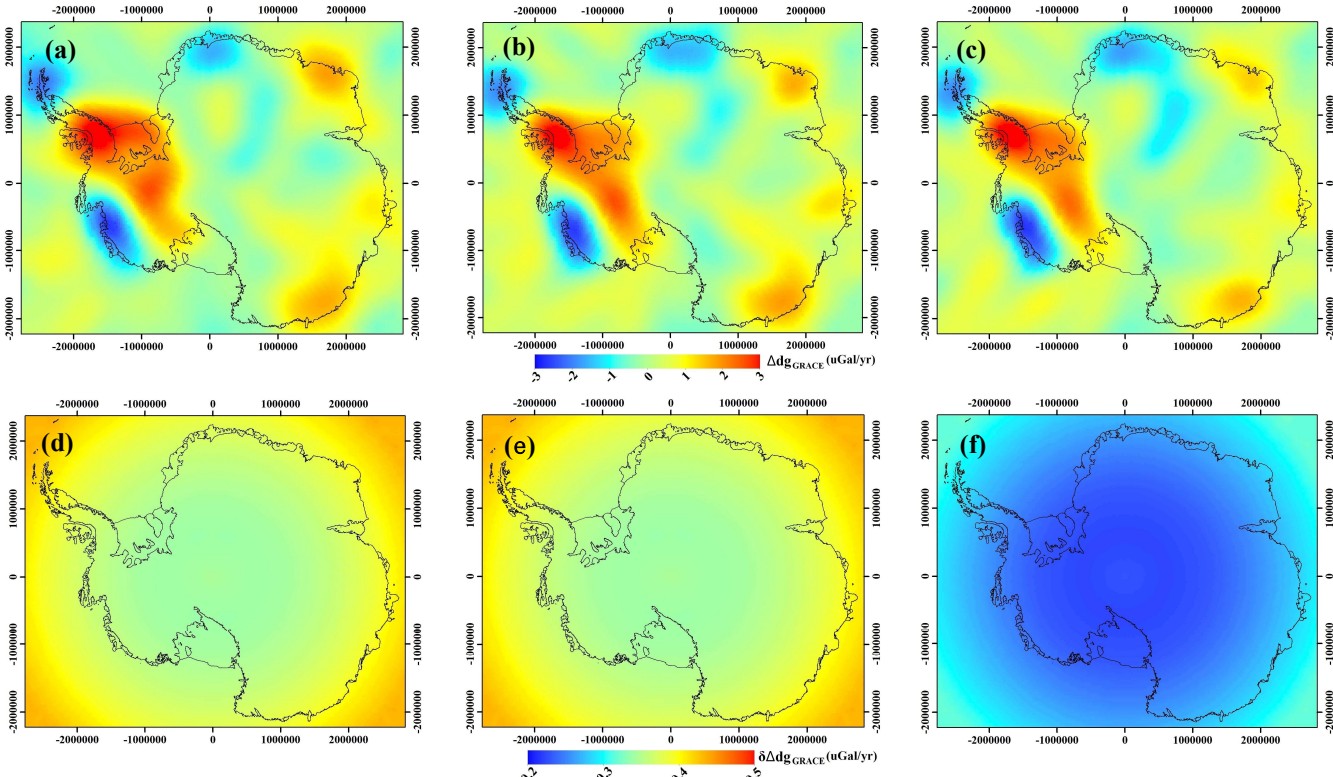

**Figure 4** Linear trend of AIS gravity variations estimated from **(a)** CSR **(b)** JPL and **(c)** GFZ solutions, and **(d)**, **(e)**, **(f)** associated uncertainties in unit of uGal/yr. These monthly solutions during 2003–2009 are consistent with the ICESat mission time.

## 3.2 Surface height variations derived from ICESat

Height variations on surface AIS were derived from Ice, Cloud, and Land Elevation Satellite (ICESat) mission, which were provided by National Snow and Ice Data Center (NSIDC), with the mission period spanning from February 2003 to October 200 2009. The object of ICESat mission is to obtain the height variations with an accuracy of ≤1.5cm/yr for spatial resolution of $100*100 km^2$ on AIS (Zwally et al., 2002). Therefore, we estimated AIS height variations by using a block crossover analysis method (Brenner et al., 2007;Gunter et al., 2009) with each block size about $100*100 km^2$. In pre-processing of ICESat data, crossover points with height variations greater than 10m/yr were deleted in order to eliminate gross errors caused by the small-scale surface roughness, undetected forward scattering and the interpolation between successive footprints, then the 3σ 205 criterion test was carried out in each block area to further eliminate the residual gross errors (Smith et al., 2005). The linear surface height variations trends were abstracted through least squares adjustment of four-parameter function consisting of constant term, linear trend, and annual periodic signals.

The inter-campaign biases (ICB) have been evaluated by several research groups through a variety of approaches, while these ICB corrections vary largely due to the selection of different reference surfaces. Zwally et al. (2015) made the correction by



using concurrent Envisat radar altimetry of the same surface in open water and thin ice in leads and polynyas in Antarctic sea
ice pack. Richter et al. (2014) and Schroeder et al. (2016) made the correction based on a near-zero surface height changes and
hydrostatic equilibrium for the snow surface above Lake Vostok and its surroundings in East Antarctica (EA) that is concluded
by repeat measurement of a GPS stake network and kinematic profile resurveys. Other methods for ICB estimations are
performed based on the assumption of near-zero elevation changes regions in EA. To date, none of ICB corrections has been
endorsed by the ICESat Science Team, NASA, or NSIDC. Therefore, uncertainties of ICB corrections also have an evident
influence on the estimation of BMB results. In this study, we used the mean ICB corrections value of Zwally et al. (2015),
Richter et al. (2014) and Schroeder et al. (2016), to avoid the artificial near-zero elevation changes assumption.

Figure 5a shows linear trend of AIS surface height variations (including ICB corrections), with the grid size of 100*100 km². 
Figure 5b shows the uncertainties of mean height variations trend. To match with the GRACE spatial resolution, a 300km
Gaussian smoothing filter was performed in the surface height variations before calculating surface gravity variations.

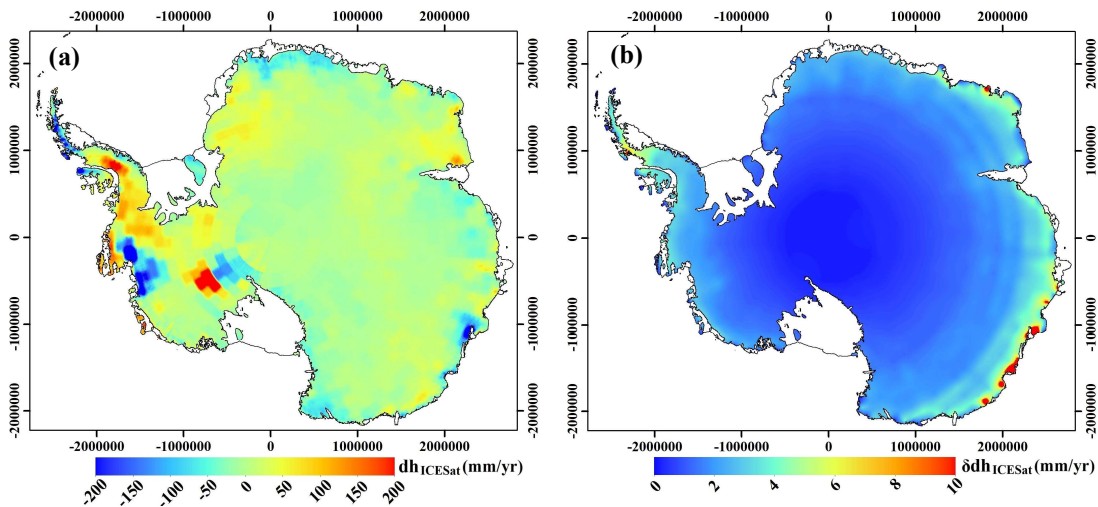

**Figure 5 (a)** Linear surface height variations trend (including ICB corrections) during February 2003 - October
2009 (truncated at 200 mm/y). **(b)** Associated uncertainties of height variation trend.

### 3.3 Firn densification model

Height variations of AIS firn layer are associated with both mass-conserving (firn compaction), mass-changing (snow
accumulation, sublimation/runoff) processes, and ice flow (Zwally et al., 2005;Gunter et al., 2014). Mass-conserving process
only influences surface height variation, and have no effect on gravity variation. Therefore, discriminating these processes is
essential for estimating volume variations from firn layer's mass-changing and ice flow, thereby contributing to converting
surface volume variations into gravity variations in combination of surface density distribution of Kaspers et al. (2004). For
this purpose, the Institute for Marine and Atmospheric Research Utrecht Firn Densification Model (IMAU-FDM) (Ligtenberg




et al., 2011) was used to account for height variations caused by firn layer's process. IMAU-FDM is presented by an improved firn densification expression that is tuned to fit depth-density observation and liquid water processes (meltwater percolation,

refreezing and retention). At AIS surface, the time-dependent IMAU-FDM is forced by the surface mass balance, surface temperature and wind speed from the regional atmospheric climate model RACMO2/ANT, to simulate the temporal evolution of firn density and surface height (Ligtenberg et al., 2011). For simplifying calculation process, this study used a constant surface firn density model from IMAU-FDM, which would cause a mean bias of -20.4kg m$^{-3}$ compared with the temporal firn density (Keenan et al., 2021), while this bias is negligible compared with the mean density of 340kg m$^{-3}$ over AIS. Figure 6a-

6b shows the linear surface height changes rate derived from IAMU-FDM and the associated uncertainties over the study period. For the consistency of spatial resolution, a 300km Gaussian filter was also performed in IAMU-FDM-derived surface height changes rate.

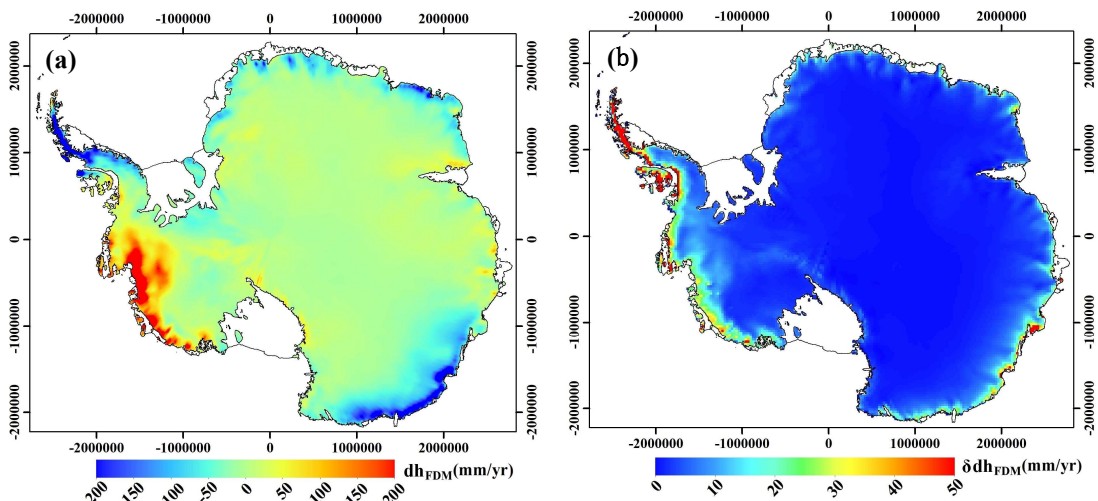

**Figure 6 (a)** Linear trend of firn layer's height variations derived from IAMU-FDM over the study period and

**(b)** associated uncertainties

### 3.4 Glacial isostatic adjustment and crustal vertical deformation rate derived from GPS

In our study, three GIA models were used to account for Earth's isostatic deformation and mass changes in the interior of the

Earth, including ICE-6G (Peltier et al., 2015;Argus et al., 2014), IJ05_R2 (Ivins et al., 2013) and W12a (Whitehouse et al., 2012). ICE-6G is a global GIA modal that is constrained to fit all available geological and geodetic observation data including GPS, ice thickness, relative sea level histories and the age of marine sedimentation. IJ05_R2 and W12a are regional GIA models that are constrained by extensive geological and glaciological data. All GIA model provides geoid rate in form of stokes coefficients, as well as predicated uplift rate in grid format. Uncertainties of ICE-6G and IJ05_R2 arise from the misfit

between uplift rates from GIA and GPS: for ICE-6G, the rms value of 0.89 mm/y is estimated from 42 GPS stations (Argus et





al., 2014); for IJ05_R2, the uniform value of 1.40 mm/y comes from 6 GPS stations with observation period over 3000 days (Ivins et al., 2013). Uncertainty of W12a is provided by the upper and lower bounds of geoid rates.

The crustal vertical deformation rate on Antarctica contains two components: Earth's isostatic deformation and load deformation of the crust due to the mass changes of ice sheet. The combined deformation rate during 1995-2013 comes from

GPS observed uplift rates from a total of 118 GPS sites listed in Sasgen et al. (2016). We used 57 of them over the study period, and removed sites with the errors greater than uplift rates. The sparse GPS data were interpolated by the GIA predicated spatial uplift distribution, to obtain the crustal vertical deformation throughout the whole Antarctic ice sheet. Figure 7a-7c shows the predicated uplift rate derived from ICE-6G, IJ05_R2 and W12a, as well as the uplift rate and their uncertainties of 57 GPS sites.

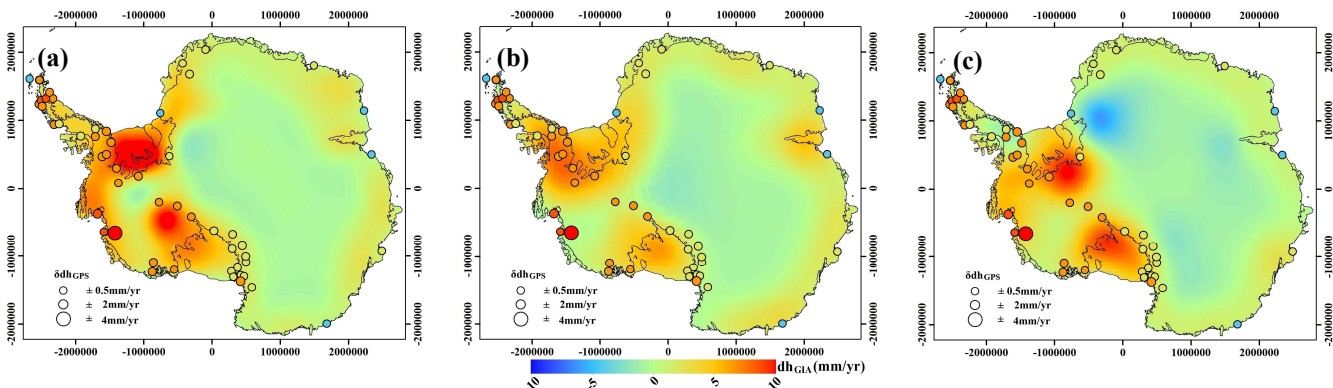

**Figure 7** GIA predicated uplift rate from **(a)** ICE-6G, **(b)** IJ05_R2, **(c)** W12a. And GPS observations for crustal vertical deformation rate and associated uncertainties during 2003-2009. The color of the circle donates the uplift rate and radius donates the uncertainties.

**3.5   Basal melting rate**

Antarctic basal melting rate was used to estimate basal water storage variations (BWSV) in combination with basal mass balance (BMB). The basal melting data used in this study is determined by Pattyn (2010), through a hybrid method that combines a priori information (such as on-site measurements data), topography, accumulation, surface temperature, geothermal heat flow data with a merged ice sheet/ice stream model. The associated uncertainty is evaluated through a series

of sensitivity experiments. The multi-source dataset used in estimating basal melting rate cover different period spanning from 1980 to 2004 (Pattyn, 2010;Van Liefferinge and Pattyn, 2013), differing from the period of this study (2003-2009). Nevertheless, this period discrepancy has little influence on our estimation of BWSV, due to the stable basal conditions caused by isolation of overlying ice sheet.

Figure 8a-8b shows the mean basal melting rate and the standard deviation for sensitivity experiments. Where the highest melt

rates are situated in West Antarctica (WA), while freezing regions are associated with subglacial mountain ranges with overlying thin ice, and the surrounding of EA. The mean basal melting rate of AIS is 5.3 mm/y, which lead to a total basal



water mass change rate of 65 Gt/y. In this study, we used a 300km Gaussian smoothing filter on the basal melting rate to match the spatial resolution of BMB.

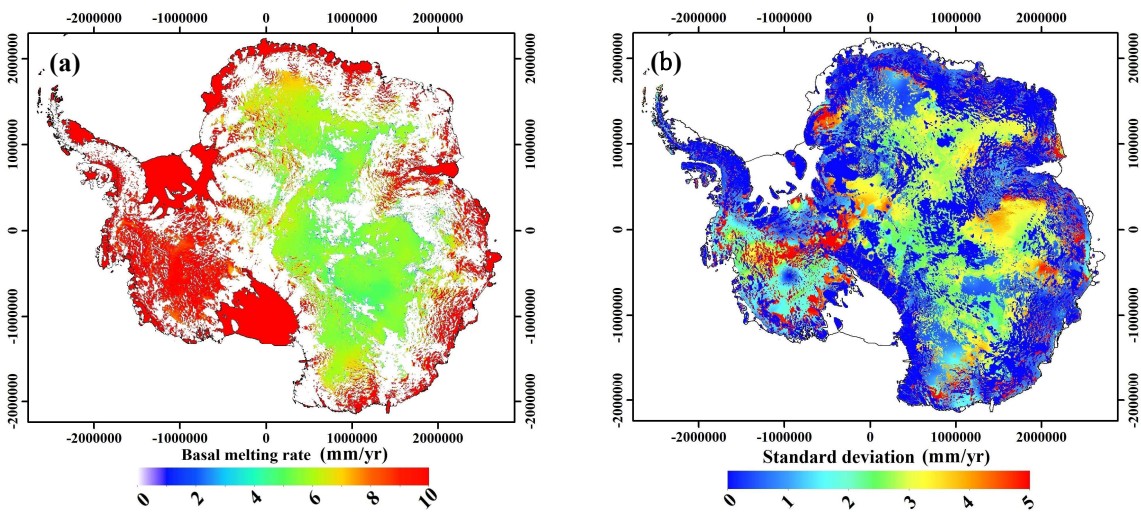

**Figure 8 (a)** Basal melting rate on Antarctic ice sheet (truncated at 10 mm/y) and **(b)** associated standard deviation. These datasets come from Pattyn (2010).

## 4.    Results and discussion

### 4.1 Basal mass balance beneath Antarctic ice sheet


Based on the gravity density forward/inversion iteration method in section 2.2, we calculated the Antarctic basal mass balance (BMB) that is mainly caused by basal water migrations. Figure 9 displays the BMB result for each iteration related to 3 glacial isostatic adjustment (GIA) models. As shown in figure 9, three BMB results all converge to stable negative value in the seventh iteration, indicating the total basal mass balance of AIS in continuously decreasing during 2003-2009. Total BMB results

related to ICE-6G and W12a show consistent greater variations in the first iteration and converge to a large value (~-20Gt/yr), while the BMB result related to IJ05_R2 shows a relatively small value.





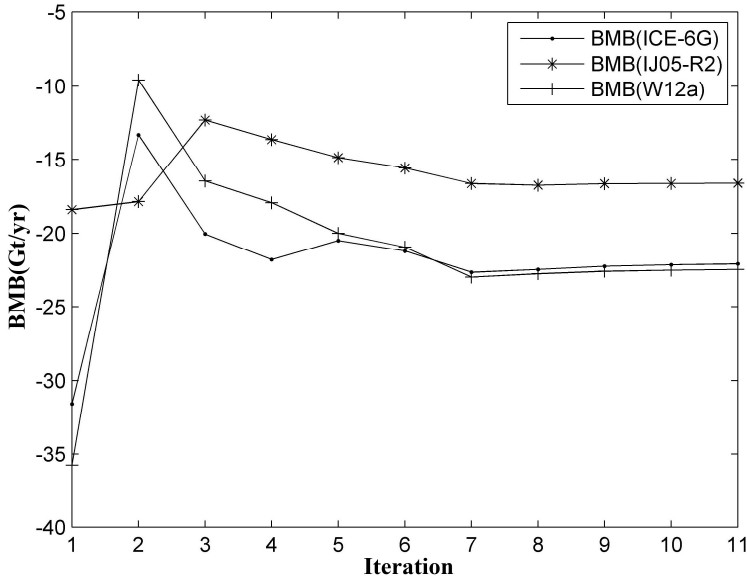

**Figure 9** Total basal mass balance (BMB) results on Antarctic ice sheet for
each iteration (in unit of Gt/yr).

Regional BMB rates related to 3 GIA models based on 18 drainage basins from Rignot et al. (2019) (shown in Figure 10a-10c) and associated standard deviations are shown in Table 1. The drainage basins division method is employed based on the spatial similarity of simulated basal meltwater pathways and surface ice flows. As shown in Table 1, although different GIA models

are used in estimating BMB, regional BMB rates related to 3 GIA models show similar values in most of the drainage basins: 3 drainage basins (B5, B6 and B9) exhibit obvious basal mass increases (with the absolute value of regional BMB rates greater than associated standard deviations and lager than 10Gt/yr), and 3 drainage basins (B2, B3 and B11) displays obvious basal mass decreases. Mean BMB rates (average of BMB rates related to 3 GIA models) in East Antarctic Ice Sheet (EAIS, including B1-B8, B17 and B18) and West Antarctic Ice Sheet (WAIS, including B9-B12 and B16) are 11 ± 12 Gt/yr and -31 ± 5 Gt/yr,

accounting for 23% and 30% of the corresponding ice-sheet mass balance estimated by satellite gravimetry method (Shepherd et al., 2018), respectively. Mean BMB rate in Antarctic Peninsula Ice Sheet (APIS, including B13-B15) is very low (-1 ± 1 Gt/yr), indicating that there are few basal mass variations. The total mean BMB rate beneath Antarctic Ice Sheet (AIS) is -21 ± 13 Gt/yr, accounting for 28% of the satellite gravimetry derived ice-sheet mass balance rate (-76Gt/yr) from Shepherd et al. (2018).

Figure 10 displays spatial distributions of BMB trend on AIS related to 3 GIA models, and associated standard deviations (STD), in form of equivalent water height (EWH). In Figure 10a-10c, red colours represent basal mass increases regions where the input volume of basal mass is greater than the output volume; blue colours represent regions with basal mass decreases; the dash regions are where the BMB rates are less than the STD, that is, BMB result is not significant in these regions. Figure 10a-10c shows similarly spatial distributions of BMB trend that related to 3 GIA models: where basal mass variations occur





mainly in WAIS, marginal regions of EAIS and Wilkes Land. The associated STD (Figure 10d-10f) also shows consistent spatial distributions: the largest STD (≥15 mm/yr) are in APIS and WAIS, which mainly comes from the STD of GRACE (about 25% of total STD estimated from Equation 11), ICESat (25%, mainly from ICB), FDM (20%) and GPS (20%); the medium STD (10-15 mm/yr) are in the marginal region of EAIS, mainly coming from GRACE (40%), ICESat (30%) and FDM (20%); the low STD (<10 mm/yr) cover a large extent the interior of EAIS, mainly coming from GRACE (45%) and

ICESat (35%).

**Table 1** Regional Antarctic basal mass balance (BMB) rates related to 3 GIA models in 18 drainage basins
and associated standard deviations, in unit of Gt/yr

| Basin | BMB(ICE-6G) | | BMB(IJ05_R2) | | BMB(W12a) | | Basin | BMB(ICE-6G) | | BMB(IJ05_R2) | | BMB(W12a) | |
|---|---|---|---|---|---|---|---|---|---|---|---|---|---|
| | Rates | Std | Rates | Std | Rates | Std | | Rates | Std | Rates | Std | Rates | Std |
| B1 | 4 | 4 | 4 | 5 | 4 | 4 | B11 | -41 | 3 | -40 | 4 | -45 | 3 |
| B2 | -10 | 4 | -11 | 4 | -11 | 3 | B12 | -1 | 1 | -1 | 1 | -1 | 1 |
| B3 | -10 | 7 | -9 | 9 | -11 | 7 | B13 | 0 | 1 | 0 | 1 | -1 | 1 |
| B4 | -1 | 4 | -1 | 5 | -1 | 4 | B14 | 1 | 1 | 1 | 1 | 1 | 1 |
| B5 | 20 | 7 | 21 | 8 | 20 | 7 | B15 | -2 | 1 | -2 | 1 | -2 | 1 |
| B6 | 12 | 4 | 11 | 5 | 12 | 4 | B16 | 4 | 4 | 2 | 5 | 0 | 4 |
| B7 | 2 | 2 | 1 | 2 | 2 | 2 | B17 | 0 | 12 | 0 | 13 | 3 | 11 |
| B8 | -6 | 9 | -8 | 10 | -4 | 8 | B18 | 0 | 2 | 0 | 2 | 0 | 2 |
| B9 | 9 | 5 | 18 | 6 | 15 | 5 | | | | | | | |
| B10 | -4 | 1 | -4 | 1 | -4 | 1 | Total | -23 | 21 | -18 | 24 | -22 | 20 |

In Figure 10a-10c, significant basal mass increases (with the BMB rates greater than associated standard deviations) occur mainly in Rockefeller Plateau (RP), Siple Coast (SC), George V Coast (GVC), Aurora Subglacial Basin (ASB), marginal area

of Dronning Maud Land (DML), Slessor Glacier (SG), Recovery Ice Stream (RIS) and Institute Ice Stream (IIS). Among them, RP, SC, GVC and ASB region possess the most obvious basal mass increases, this is attributed to the low-lying basal terrain facilitating the accumulation of basal meltwater driven by basal hydraulic potential gradient (Fig 18, (Göeller, 2014)), and Wright et al. (2012) revealed the possibility of basal meltwater accumulation in ASB region by revealing extensively distributed basal pressure melting through regional airborne geophysical survey and numerical ice sheet modelling method. In

SC region, the mechanism of basal meltwater accumulation is also revealed in MacAyeal and Whillans ice streams, where the similar basal hydrological systems are also found (Fricker et al., 2016;Gray et al., 2005); additionally, the accumulated basal meltwater in SC region might also partially account for the dynamic thickening of Kamb ice stream. Similarly, basal mass increases in IIS regions should be also caused by low basal hydraulic potential and low-lying basal terrain that conduces to the collection of surrounding waters. Basal mass increases in RIS region arise from subglacial lake water accretion and discharging

into the bedrock trench underneath the Recovery Glaciers (Bell et al., 2007), and similar basal mass increases might also occur in SG region. In DML region, basal mass increases occur in basal ridge, which might be attributed to the basal supercooling

condition in DML that conduce to the basal meltwater refreezing in basal ridge when flowing upward driven by basal hydraulic gradient. The possibility of basal water flowing upward to basal ridges has been verified in Gamburtsev Subglacial Mountain (GSM) through analysing a comprehensive geophysical data set (Creyts et al., 2014), and the consistent pattern is also shown

in our BMB result, although the pattern is not significant enough in GSM region.

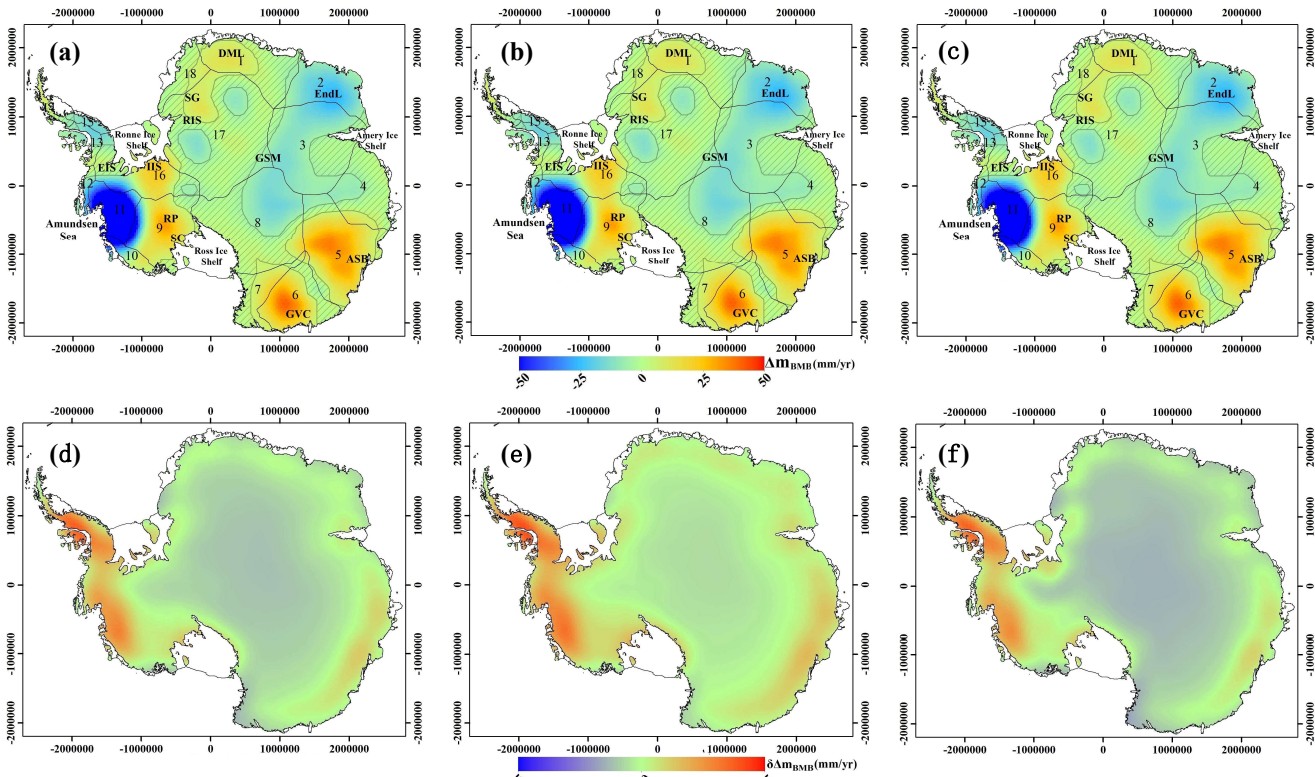

**Figure 10** Basal mass balance (BMB) rates related to **(a)** ICE-6G, **(b)** IJ05_R2 and **(c)** W12a in unit of mm/yr, and **(d) (e) (f)** associated standard deviations during 2003-2009. AIS drainage basins in **(a) (b) (c)** are determined according to the definitions of Rignot et al. (2019), and the shaded area are regions where the BMB rates less than the associated standard deviations.

SC = Siple Coast; DML = Dronning Maud Land; EndL = Enderby Land; ASB = Aurora Subglacial Basin;

IIS = Institute Ice Stream; GVC = George V Coast; RIS = Recovery Ice Stream;

SG = Slessor Glacier; RP = Rockefeller Plateau; GSM = Gamburtsev Subglacial Mountain

Significant basal mass decreases occur mainly in Amundsen Sea coast, the interior of EAIS and Enderby Land (EndL) region. Among them, the most significant basal mass decreases along Amundsen Sea coast region are caused by the active ice melting in response to basal geothermal flux (Schroeder et al., 2014), this lead to a majority of basal water discharging into Amundsen Sea through basal channels (Gray et al., 2005). Basal mass decreases in the interior of EAIS are caused by the basal meltwater

generated from basal pressure-melting (Augustin et al., 2007) flows to marginal regions, driven by basal hydraulic potential gradient (Göeller, 2014). The decreases in EndL region lacks justification, which might also be caused by the volume scattering



in satellite altimetry data that underestimate surface height variation, or caused by the error in FDM that lead to the underestimation in surface mass loss in gravity forward modelling process, thereby impeding BMB estimation in these regions, and verifying these possible conjectures need further effort.

## 4.2 Basal water storage variations beneath Antarctic ice sheet

Antarctic basal water storage variations (BWSV) were estimated based on BMB and Antarctic basal melting rate data, through the combination method in section 2.2. It reveals mass variations of liquid water storage beneath Antarctic ice sheet. Table 2 displays regional BWSV rates related to 3 GIA models and associated standard deviations. As shown in Table 2, regional BWSV rates related to different GIA models also show similar values in each drainage basin: 2 drainage basins (B5 and B6) exhibit obvious basal water increases, and 1 drainage basin (B11) displays significant basal water decreases. Mean BWSV (average of BWSV rates related to 3 GIA models) rates in EAIS, WAIS are $47 \pm 12$ Gt/yr and $-4 \pm 5$ Gt/yr, and no obvious BWSV occurs in APIS. The total mean BWSV rate beneath AIS is $43 \pm 13$ Gt/yr, accounting for 66% of the basal melting rate (65Gt/yr) (Pattyn, 2010), which indicates most of the basal melted water is stored in ice-bed interface.

**Table 2** Regional Antarctic basal water storage variations (BWSV) rates and associated standard deviations related to 3 GIA models in 18 drainage basins, in unit of Gt/yr

| Basin | BWSV(ICE-6G) | | BWSV(IJ05_R2) | | BWSV(W12a) | | Basin | BWSV(ICE-6G) | | BWSV(IJ05_R2) | | BWSV(W12a) | |
|---|---|---|---|---|---|---|---|---|---|---|---|---|---|
| | Rates | Std | Rates | Std | Rates | Std | | Rates | Std | Rates | Std | Rates | Std |
| B1 | 5 | 4 | 4 | 5 | 4 | 4 | B11 | -16 | 4 | -15 | 4 | -15 | 4 |
| B2 | -1 | 4 | -2 | 4 | -1 | 3 | B12 | 0 | 1 | 0 | 1 | 0 | 1 |
| B3 | -2 | 8 | -2 | 9 | -2 | 7 | B13 | 0 | 1 | 0 | 1 | 0 | 1 |
| B4 | 4 | 5 | 4 | 5 | 4 | 4 | B14 | 0 | 1 | 0 | 1 | 0 | 1 |
| B5 | 17 | 7 | 18 | 8 | 15 | 7 | B15 | 0 | 1 | 0 | 1 | 0 | 1 |
| B6 | 11 | 4 | 11 | 5 | 10 | 4 | B16 | 0 | 5 | 4 | 5 | 2 | 5 |
| B7 | 4 | 2 | 4 | 2 | 4 | 2 | B17 | 6 | 12 | 6 | 14 | 7 | 11 |
| B8 | 2 | 9 | 1 | 10 | 3 | 8 | B18 | 2 | 2 | 2 | 2 | 2 | 2 |
| B9 | 6 | 5 | 13 | 6 | 10 | 5 | | | | | | | |
| B10 | 0 | 1 | 0 | 1 | 0 | 1 | Total | 38 | 22 | 48 | 25 | 43 | 21 |

Figure 11a-11f displays the spatial distributions of BWSV rates on AIS related to 3 GIA models, and associated uncertainties, in form of equivalent water height (EWH). Where red colours represent basal water increases that arise from basal water migration and basal melting; blue colours represent basal water decreases that caused by basal water runoff or basal water refreezing; blue dots represent locations of active subglacial lakes that observed through surface height variations (Smith et al., 2009), grey dots are locations of definite or fuzzy subglacial lakes that detected by radio-echo sounding (RES) measurement



(Andrew and Martin, 2012). The spatial distributions of BWSV shown in Figure 11a-11c is similar to that of BMB in Figure 10a-10c, and the difference situated mainly in marginal regions of EAIS, where basal water increases are more extensive.

Results also show similar spatial distributions between obvious basal water increases (with the BWSV rates greater than associated standard deviations) and active subglacial lakes (blue circles in Figure 11a), such as the active subglacial lakes in

IIS, RP, GVC, ASB and SG regions. This indicates that the basal water volumes in most active subglacial lakes are increasing, despite water drainage occurs frequently; among them, the most typical justification is available in RP region, where the continued surface height increases in most subglacial lakes are found (Siegfried and Fricker, 2018). Definite or fuzzy lakes (grey circles in Figure 11b) are distributed mainly in regions of EA with low BWSV rates, exceptions are Concordia Subglacial lakes in Dome C (DC) where the basal water volumes are obviously increasing, this is attributed to the enhanced basal melting

in Concordia Ridge, Concordia Subglacial Lakes and Vincennes Basin regions that conduce to the replenishment of basal water storage (Carter, 2007).

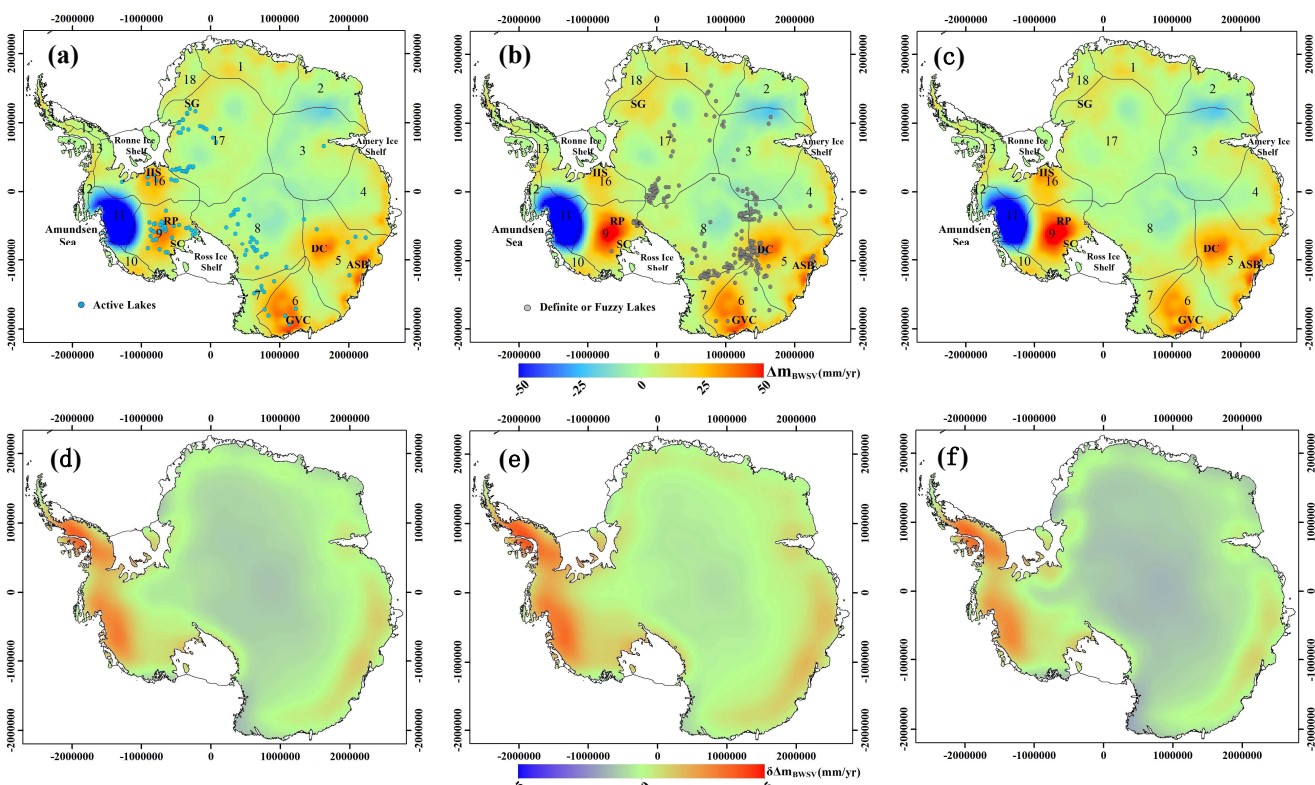

**Figure 11** Basal water storage variations (BWSV) rates related to **(a)** ICE-6G, **(b)** IJ05_R2 and **(c)** W12a in unit of mm/yr, and **(d)** **(e) (f)** associated standard deviations during 2003-2009. Blue dots represent the location of active subglacial lakes observed by altimetry method, and grey dots represent the location of definite or fuzzy subglacial lakes detected by RES method.  AIS drainage basins are determined according to the definitions of Rignot et al. (2019). DC = Dome C.



In most regions of AIS, obvious basal water increases appear at regions with rapid ice flows, such as Bindschadler, MacAyeal and Whillans ice streams in SC region, Ninnis and Mertz ice streams in GVC region, Totten ice stream in ASB region and Institute ice streams in IIS region. Table 3 listed the locations, names and velocities of some fast ice streams over AIS, and

corresponding basal water variations estimated in this study. Among them, most flow velocities are increasing during the study period. For example, flow velocities of Bindschadler and MacAyeal ice streams were accelerating during 2001-2014, which is attributed to the instability in the subglacial till and subglacial water flow (Hulbe et al., 2016); the acceleration of Ninnis and Mertz ice streams arise from the enhanced basal slip (Rignot et al., 2011); flow velocity of Totten ice stream had increased by 18% during 2000-2007 (Li et al., 2016). Regions without obvious basal water variations display no obvious flow velocity

increases, such as flow velocities in Rutford ice stream (Gudmundsson and Jenkins, 2009) in WA, and Fischer, Mellor and Lambert ice streams along Amery ice shelf. The spatial distribution relations between obvious basal water increases and rapid/accelerated flow velocities in most regions of AIS indicates that the accumulated basal water contribute to lubricating ice-rock interface and promote the ice flow above. However, exceptional cases are situated along Amundsen Sea coast (Pine Island and Thwaites glaciers), where significant basal water decreases follow rapid/accelerated ice flow (Han et al., 2016;Kim

et al., 2015). In these exceptional cases, rapid/accelerated flows arise from the combined effect of high basal geothermal flux induced basal ice-rock meltwater lubrication (Gray et al., 2005;Schroeder et al., 2014), as well as the buttressing loss induced by the extension of the shear zone and progressive disintegration (Kim et al., 2015). In summary, we believe that in most regions of AIS (except the Amundsen Sea coast), basal water storage variations appear to have an important effect on ice flow, except for other factors such as the stiffness of ice shelf, and flow velocities will further accelerate if basal water continues to

increase in these regions.





**Table 3** Locations, names and flow variations of over AIS and corresponding BWSV. Columns give (1) locations of ice stream that shown in Figure 11; (2) ice stream names; (3) flow velocities, R donate rapid ice flows, S donate slow ice flows less than 500m/yr; (4) flow variations, ↑/N/↓ donate accelerated/steaty/slowed down flow; (5) basal water storage variations, ↑/N/↓ donate obvious increased/steady/decreased basal water storage; (6) reference.

| Locations | Names | Flow velocities | Flow variations | BWSV | Reference |
|---|---|---|---|---|---|
| SC | Bindschadler | R | ↑ | ↑ | (Hulbe et al., 2016) |
| | MacAyeal | R | ↑ | ↑ | |
| | Whillans | R | ↓ | ↑ | (Winberry et al., 2014) |
| GVC | Ninnis | R | ↑ | ↑ | (Rignot et al., 2011) |
| | Mertz | R | ↑ | ↑ | |
| ASB | Totten | R | ↑ | ↑ | (Li et al., 2016) |
| Along Amery ice shelf | Fischer Mellor Lambert | R | N | N | (Pittard et al., 2015) |
| WA | Rutford | R | N | N | (Gudmundsson and Jenkins, 2009) |
| IIS | Institute | R | ⊗ | ↑ | (Rignot et al., 2011) |
| Amundsen Sea coast | Pine Island | R | ↑ | ↓ | (Han et al., 2016); |
| | Thwaites | R | ↑ | ↓ | (Kim et al., 2015); |

## 5. Conclusions

In this study, we presented a layered gravity density forward/inversion method for estimating basal mass balance (BMB) and basal water storage variations (BWSV) rates beneath AIS during 2003-2009, by a combination of multi-source satellite observation data including satellite gravity, altimetry and GPS data, and relevant models including firn densification model (FDM), glacial isostatic adjustment and basal melting rate. The presented method used an iteration approach for separating gravity components caused by surface firn process and ice flow from the integrated gravity variations, to obtain the gravity variations and mass changes caused by BMB. When estimating BMB rates, 3 monthly GRACE gravity field solutions (CSR, GFZ and JPL), 3 inter-campaign biases corrections and 3 GIA models were used to obtain the average gravity variations, height variations and Earth's isostatic adjustment, to reduce possible errors.

Results reveal obvious spatial variability of BMB during 2003-2009, showing the potential for detecting BMB rates beneath AIS through the layered gravity density forward/inversion method. Significant basal mass decreases located mainly along Amundsen Sea coast, the interior of EAIS and Enderby Land region, while basal mass increases situated mainly in Rockefeller





Plateau, Siple Coast, Institute Ice Stream regions and marginal of EAIS. Total BMB rates beneath AIS are decreasing with the mean rate of -21 ± 13 Gt/yr (EAIS: 11 ± 12 Gt/yr, WAIS+APIS: -32 ± 5 Gt/yr), accounting for 28% of the mass balance rate

(-76Gt/yr) from Shepherd et al. (2018); and this indicates that the basal water migration between basal drainage systems and oceans is non-negligible in estimating AIS mass changes. BWSV rates exhibit a similar spatial distribution as BMB, and the total BWSV rates are increasing with the rate of 43 ± 13 Gt/yr (EAIS: 47 ± 12 Gt/yr, WAIS+APIS: -4 ± 5 Gt/yr), which is attributed to the active basal melting rate that producing sufficient meltwater to replenish the basal meltwater loss, and the excessed meltwater is stored beneath AIS that bring about increase in basal meltwater storage. BWSV results also show similar

spatial distribution between significant basal water increases and active subglacial lakes, indicating that the basal water in most active subglacial lakes is increasing, despite water drainage occurring frequently. Basal water in most definite or fuzzy lakes is relatively stable, exceptions are Concordia Subglacial lakes where the basal water is obviously increasing. We also found the spatial distribution relations between obvious basal water increases and rapid/accelerated flow velocities in most of the AIS (except the Amundsen Sea coast region), which indicates that basal water storage variations appear to have an important

effect on ice flow, and flow velocities will further accelerate if basal water continues to increase in these regions.

The main errors in estimating BMB come from the uncertainty of GRACE, FDM and inter-campaign biases of altimetry data, especially the current results of different inter-campaign biases corrections remain large discrepancy. A more elaborate result is desirable if utilizing more reliable and higher spatial/temporal resolution data, which deserves further studies.



## Code/Data availability

BMB and BWSV data is available at https://github.com/Kangjingyu17/BMB-BWSV.git

Dataset used in this study is listed as follows table.

| Data set | URL | last access |
|---|---|---|
| GRACE | ftp://isdcftp.gfz-potsdam.de/grace/Level-2/ | 30 August 2021 |
| ICESat | https://nsidc.org/data/icesat/ | 30 August 2021 |
| BEDMAP2 | https://secure.antarctica.ac.uk/data/bedmap2/ | 30 August 2021 |
| GPS | https://dep1doc.gfz-potsdam.de/documents/102 | 30 August 2021 |
| ICE-6G | https://www.atmosp.physics.utoronto.ca/~peltier/data.php | 30 August 2021 |
| W12a | http://www.pippawhitehouse.com/ | 30 August 2021 |

Any other specific data and code of this study is available on request to the authors.

## Author contributions

Kang and Lu conceived the research. Kang performed forward modelling and inversion calculation. Shi processed ICESat data. Kang, Li and Zhang collected the verification data. Kang, Lu and Li interpreted the Antarctic basal mass/water storage variation, Kang, Lu and Zhang wrote the manuscript. All authors read and commented on the manuscript.

## Competing interests

The authors declare that they have no conflict of interests.

## Acknowledgements

The author would like to thank Regional glacial isostatic adjustment and CryoSat elevation rate corrections in Antarctica (REGINA), Dr Peltier and Dr Whitehouse for the distribution of GPS, ICE-6G and W12a data respectively. We are also grateful to Dr Ligtenberg, Dr Pattyn, Dr Van Liefferinge and Dr Ivins, for the use of FDM/firn density data, basal melting rate data and IJ05_R2 data.

## Financial support

This research was jointly funded by the Natural Science Foundation of China (Grant No. 41674085 and 41874093).



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
