# Peer review of "Basal Water Storage Variations beneath Antarctic Ice Sheet Inferred from Multi-source Satellite Data"

_The Cryosphere, 2021_

## Referee Comment (RC2)

The manuscript aims to analyze the discrepancies between different observation methods of the Antarctic Ice Sheet (AIS) by attributing these discrepancies to basal melt water. This is a relevant and very interesting topic that is complicated by the number of observations and models that need to be included. Unfortunately, the forward gravity problem is not adequately described in the manuscript, making it impossible to evaluate the soundness of the presented method.

In section 2.3, it is stated that the forward gravity response can be computed by dividing the surface of the AIS into oblique triangular prisms from which a gravity response can be computed using Eq. (6). However, this equation does not show how the integral is performed. To define an oblique triangular prism, the three top corner points and bottom corner points must be specified (oblique means that the top is not directly over the bottom). Moreover, it should also be described how the sides of the prisms are defined, i.e. are these 2D planes following curved lines on the sphere or are the "flat" in some local projection. Eventually, Eq. (6) should be replaced by some closed form expression describing how the gravity response is computed from the six corner points.

In section 3, each of the data sets and models are introduced in terms of height changes related to the components in Eq. (2). It is not described how the change in gravity is computed from height changes, i.e. how are the components in Eq. (1) computed from the components in Eq. (2). Although not stated, this must somehow be related to Eq. (6) in section 2.3. However, this relation is not trivial since Eq. (6) represents the gravity response from a prism element of constant density. The computed response must be a *change in gravity with time* and as the authors state there are several processing going on, such as compaction, height changes, vertical displacement, etc. A gravity change indicates values with respect to some reference – what is this reference field?

This was a short explanation of why it is difficult to assess the methodology of the manuscript. It is simply not properly described what has been done with the data. In the following, I have some more in depth comments, but first I would also like to stress out one issue that possibly makes the results invalid. In section 3.1, Eq. (13) describes how changes in gravity are computed from the GRACE spherical harmonic coefficients. In this expression, the change in gravity is evaluated on the surface of a sphere of radius R (the actual radius is not stated, but I assume it is the same as in the GRACE coefficient file, otherwise the coefficients would have to be scaled). Similarly, the gravity response of the forward model must be computed at some point in space (this computation surface is not specified in the manuscript – it should be!). Now, the GRACE satellite observes mass changes BELOW the satellite. However, GRACE is evaluated on a spherical surface which is (generally) sometimes below and sometimes above the origin of the mass changes. In principle Eq. (13) is not valid below the surface of the Earth, but it can be argued that this is just some kind of harmonic downward continuation. Now, imagine first a mass loss below the spherical surface. The GRACE satellite would observe less mass and this a decrease in gravity. Similarly, using the forward model, a decrease in gravity would be computed from a mass loss below the computation point. However, imagine the mass loss occurs above the spherical surface. Again, GRACE measures at satellite altitude and would observe a decrease in gravity. On the other hand, using the forward model, a mass decrease above the computation point would lead to an increase in gravity. This results in an inconsistency which may make the entire inversion process invalid. For consistency, the computation surface should be the same for both GRACE and forward model AND it should be somewhere above the surface of the ice sheet.

For these reasons, I recommend to reject the manuscript. However, I do find the subject of the manuscript quite relevant and would very much like to see the study published after a thorough workover.

**Section 2.1**

Line 55: "Earth's load deformation" – is this the elastic deformation of the crust due to changes in the (ice) load?

Line 65: "BMB and basal melting/refreezing also influences ice sheet's vertical movement by controlling basal mass changes and ice-water phase change process" – I do not understand the link? Especially since the model applied later assumes a layer of constant thickness.

"$\Delta dg_{GRACE}$ are Antarctic integrated time-variable gravity variations (including load effect)…" – Later in Eq. (13) the load effect is removed using load Love numbers?

**Section 2.3**

I have never seen the use of *oblique* triangular prisms in gravity computations, but I am not completely up to date on this issue. However, for the oblique prism, the top is not directly over the bottom – in Figure 3 one set of vertices ("corners") are defined – are those the top or the bottom?

It is not described how the triangular prism are defined – the illustration in Figure 3 is not sufficient. In general, six points are required to define the corners, but also the surfaces must be defined. Are they "flat" surfaces in some map projection or are they defined by planes on the sphere? More specifically it should be described how the integral in Eq. (5) is carried out.

Eq. (5) is presented with a reference to the book by Hofmann-Wellenhof and Moritz. Although the author guidelines do not state this specifically, I find it pointless to cite a 400-page book without specifying where in the book this one equation is found.

In this same book, eq. 1-12 gives the gravitational potential from an arbitrary volume with mass $\rho$ as

$$V(\pmb{x}_P) = G\rho \iiint \frac{1}{|\pmb{x}_Q - \pmb{x}_P|} d^3v$$

where $\pmb{x}_P$ is the computation point (where gravity is computed) and $\pmb{x}_Q$ is the evaluation point (integration point of prism). In the manuscript, the coordinates $(x, y, z)$ are not defined. However, comparing eq. (5) with the above would indicate that

$$\pmb{x}_Q - \pmb{x}_P = (x, y, z)$$

meaning that the reference frame origin corresponds to the computation point. In my opinion Eq. (5) thus introduces some unnecessary confusion, which can be handled quite easily by notation. As an example:

$$dg_p \equiv dg(x_p, y_p, z_p) = -G\rho \iiint \frac{\Delta z}{(\Delta x^2 + \Delta y^2 + \Delta z^2)^{\frac{3}{2}}} dx \, dy \, dz$$

with e.g. $\Delta x = x - x_p$ , indicating that $dg_p$ is the gravity variation at a single point in space and also that the coordinates are relative to this point.

Since gravitational attraction is the gradient of potential, eq. (5) is an approximation using only the vertical derivative of the above expression– for this reason I guess the authors restrict themselves to the area directly below the computation point (i.e. 500 km radius stated in line 140). Using all three gravity components and a larger surface area could possibly add more information at the cost of increased computational burden. However, since GRACE products usually come in the form of spherical harmonic coefficients, one could also simply compute the potential.

The computation coordinates must also be specified. Eq. (13) indicates that GRACE it evaluated on the sphere – the prism response should be computed at the same location.

There should be a reference to eq. (6) or the authors should show how it is derived.

In Eq. (8) I do not see how the units add up? Density = density x height? Possibly this should be divided by the fixed height of the layer?

It is not until Eq. (7) that is becomes clear what "gravity variations" refer to. Eqs. (5-6) are not "gravity variations" since a reference model is not yet specified. In these expressions, I recommend replacing $dg$ with $g$.

As the authors state, mass can be added below the ice sheet, above the ice sheet and the entire column can be shifted up or down, all contributing to mass variation. In Eq. (7), it is indirectly stated that mass variation is attributed to changes in density of the thin-layer of constant thickness below the ice sheet. The reference model is thus a thin layer of water. This should be more clearly stated.

As I understand, mass variations are also attributed to surface processes and vertical displacement of the ice sheet, but these variations are computed via a forward model and are not allowed to vary during the inversion iteration process.

**Section 3**

It is my impression that data used in the study represent a linear trend corresponding to the period from 2003 to 2009. The objective in the following sections is therefore to derive such a linear trend for each dataset. This should be clearly stated.

**Section 3.1**

The expression in Eq. (13) is the radial derivative of the potential, which is in accordance with the approximation used in Eq. (5). The horizontal components are neglected. (However, again the potential is easily evaluated without approximation).

Furthermore, Eq. (13) is evaluated on the surface of the sphere. Strictly speaking, this expression is not valid below the attracting surface of the Earth. Basically the satellite observes masses below and therefore a downward attraction. In the case presented here, some mass (or mass changes) will be situated above the sphere resulting in an upward pull. For the forward prism model to be consistent with GRACE observations in this case, you would have to compute the prism response on some surface above the masses and then harmonically continue this surface downward to the sphere.

I would recommend instead to compute the response on some surface above the attractive masses, e.g. at GRACE satellite altitude.

Additionally, there might be some issues computing the gravitational response inside the attracting mass of the triangular prisms, but this depends on the formulas.

The introduction of load Love numbers means that the indirect response from (elastic) crustal load deformation is removed from the GRACE signal. This should be explained.

It is also stated in line 190 that modified harmonic coefficients are used. How are these modified? What are the reference coefficients? These reference coefficients are also related to the reference model of the forward computation.

**Section 3.4**

I do not understand how the GPS stations are used. The process is described as "The sparse GPS data were interpolated by the GIA predicated spatial uplift distribution, to obtain the crustal vertical deformation throughout the whole Antarctic ice sheet".

Does this mean that the crustal (elastic?) deformation is estimated as residuals between GNSS observations and GIA model, extrapolated over the area? Or is the GIA model somehow extended?

Additionally, the crustal deformation signal is already removed from the GRACE observations in Eq. (13) using load Love numbers. If you additionally compute this response in the forward model, you may be accounting for this signal twice.

To summarize, the sub-sections in section 3 gives the following variables:

Section 3.1 – GRACE:                     $dg_{GRACE}$

Section 3.2 – ICESat:                      $dh_{ICESat}$

Section 3.3 – Firn densification model:  $dh_{FL} + dh_{IF}$

Section 3.4 – GIA + GNSS:              $dh_{IVM} + dh_{CVD}$

Section 3.5 – Basal melt rate:          $dh_{IVM}$ (or are these only mass changes?)

How is the gravity variation computed from all these different components? I.e. how is $dg_{surf}$, $dg_{GIA}$ and $dg_{IVM}$ computed from the trends presented in section 3?